# Six ‘Must-Have’ Minerals for Life’s Emergence: Olivine, Pyrrhotite, Bridgmanite, Serpentine, Fougerite and Mackinawite

**DOI:** 10.3390/life10110291

**Published:** 2020-11-19

**Authors:** Michael J. Russell, Adrian Ponce

**Affiliations:** 1Dipartimento di Chimica, Università degli Studi di Torino, via P. Giuria 7, 10125 Turin, Italy; 2Jet Propulsion Laboratory, California Institute of Technology, Pasadena, CA 91109, USA; adrian.ponce@jpl.nasa.gov

**Keywords:** astrobiology, Hadean, carbonic ocean, proton gradient, redox gradient, solar system, submarine alkaline vents, emergence of life, exoplanets

## Abstract

Life cannot emerge on a planet or moon without the appropriate electrochemical disequilibria and the minerals that mediate energy-dissipative processes. Here, it is argued that four minerals, olivine ([Mg>Fe]_2_SiO_4_), bridgmanite ([Mg,Fe]SiO_3_), serpentine ([Mg,Fe,]_2-3_Si_2_O_5_[OH)]_4_), and pyrrhotite (Fe_(1−x)_S), are an essential requirement in planetary bodies to produce such disequilibria and, thereby, life. Yet only two minerals, fougerite ([Fe^2+^_6x_Fe^3+^_6(x−1)_O_12_H_2(7−3x)_]^2+^·[(CO^2−^)·3H_2_O]^2−^) and mackinawite (Fe[Ni]S), are vital—comprising precipitate membranes—as initial “free energy” conductors and converters of such disequilibria, i.e., as the initiators of a CO_2_-reducing metabolism. The fact that wet and rocky bodies in the solar system much smaller than Earth or Venus do not reach the internal pressure (≥23 GPa) requirements in their mantles sufficient for producing bridgmanite and, therefore, are too reduced to stabilize and emit CO_2_—the staple of life—may explain the apparent absence or negligible concentrations of that gas on these bodies, and thereby serves as a constraint in the search for extraterrestrial life. The astrobiological challenge then is to search for worlds that (i) are large enough to generate internal pressures such as to produce bridgmanite or (ii) boast electron acceptors, including imported CO_2_, from extraterrestrial sources in their hydrospheres.


*… molecular physics is the true basis of biology. [1]*



*… physics approximates biology because there is no such thing as an organism at thermodynamic equilibrium. [2]*


## 1. Introduction

Life being an emergent far-from-equilibrium, actively dynamic structure suggests that astrobiological exploration should focus on wet worlds where mantle convection, coupled to hydrothermal convection, resulted in strong electrochemical disequilibria commensurate with life and its emergence [3,4,5,6,7,8,9,10]. It is not enough to base astrobiological exploration on the detection of wafts of organic molecules that, after all, merely hint at the presence of reduced carbon. Indeed, most planets and moons in our outer solar system were accreted from native iron meteorites and carbonaceous chondrites (CCs), and consequently likely amassed hydrocarbon fuel aplenty as observed on Titan and possibly Enceladus [11,12,13]. What often seems to be missing is the other pole for the ‘battery’—the oxidants or electron acceptors [14].

A fundamental understanding of what drove life into being on our planet requires knowledge of how inorganic minerals, accreted from the solar disc, responded to physicochemical pressure and stresses induced by convection and differentiation in an early magma ocean [7,8,15,16]. These dynamic processes provided the ‘bio-elements’ C, H, O, N, P, S, the transition metal sulfide and oxide clusters and both the reductants and oxidants necessary for life’s emergence [17,18,19,20,21,22]. Since the primordial hydrogen atmosphere was blown beyond the snow line by the intense solar wind [23], volatiles degassing from the mantle would have included carbon dioxide, nitrogen and water [24,25]. Minor concentrations of oxidants such as nitrogen oxides (NOx) were generated from these gases, mainly through the action of cloud-to cloud-lightning in the atmosphere and occasional coronal mass injections reaching several GeV—the NOx thenceforth to be rained into the early ocean [26,27,28,29].

One fuel necessary for life’s emergence on Earth was H_2_—or the electrons prized therefrom—degassing from heterogeneous sources deep in the mantle and at the core-mantle boundary [30,31,32,33]. This hydrogen would have been joined by formate and methane, reduced from CO_2_ abiotically in the ~40 to 80 km thick Hadean ocean crust [34,35,36,37,38,39]. Apart from the ‘electricity’ provided by the oxidation of hydrogen, another driving force that couples with redox is the ‘proticity’ provided by a proton gradient [3,40]. Together, these are the reductants and oxidants and gradients responsible for life’s emergence as detailed in the submarine alkaline vent theory (AVT) (Figure 1 and Figure 2) [4,41,42,43]—a theory receiving a recent endorsement from the microfluidic experiments of Hudson and his collaborators involving nickel in an Fe(Ni)S barrier as catalyst as well as a steep proton gradient [44].

In order to produce these fuels, redox and pH gradients that mediated life’s emergence required just six ‘must-have’ minerals (Figure 1) [42,62]. Our study brings to the fore the centrality of physical and mechanical (i.e., conformational) responses of some of these minerals as disequilibrium-converting engines, or as components thereof [2,9,14,18,37,63,64,65,66,67]. For, as we learnt from the long history of continental drift debate, any theory devoid of mechanisms, i.e., engines, is doomed to a short life-span [18,62,68]. These six minerals are divided into two groups:The first group contains the four minerals setting the stage for the geochemical disequilibria that drives life into being through the macro-engines of convection, to give rise to a CO_2_-rich, transition-metal bearing acidulous Hadean ocean [39,46].The second group contains the two mineral precipitates—ferrous iron-rich oxyhydroxides and monosulfides—constituting a membrane that keeps the two contrasting fluids from immediate mixing. In AVT the oxyhydroxides are the disequilibria-converting nano-engines and solid electrolytes, while the monosulfides are the electron conductors and proto-hydrogenases. Combined, they have the potential to mediate the disequilibria and bring life into being (Figure 2) [9].

## 2. Four Minerals to Set the Stage for Life’s Emergence

The four essential minerals in planetary bodies needed to produce the chemical disequilibria and materials required for life’s emergence are olivine ([Mg>Fe]_2_SiO_4_), bridgmanite ([Mg,Fe]SiO_3_), serpentine ([Mg,Fe,]_2-3_Si_2_O_5_[OH)]_4_) and pyrrhotite (Fe_(1-x)_S). We begin with olivine, a ubiquitous mineral contributing to the primary extrusive and intrusive basaltic or komatiitic ocean floor, crust and upper mantle (originally as short-lived magma oceans [70]) to a myriad of putative planets—the precursor to suites of minerals produced through the stressors of pressure, thermal gradients and hydration. We proceed with the most abundant of these—bridgmanite (Mg,Fe)SiO_3_)—which is hidden at depth out of reach of spectroscopy. Its occurrence must be gauged experimentally from a planet’s metamorphic response to size and dehydration and as supported by seismic profiles [15,71]. Hydration of olivine makes for the alkaline nature of the feeders to life in the AVT, while pyrrhotite is the dominant abiogenic source of the sulfur [39,44,46]. Olivine and pyrrhotite react with hydrothermal solutions at ≤150 °C to produce the serpentine mineral and the alkaline fluids bearing ≤10 millimoles of hydroxyl and ≤10 micromoles of bisulfide [39,56,72]. The result is a strong pH disequilibrium between the alkaline fluid (pH ~11 to 11.5) and the relatively low, though varying, pH of the early ocean (~pH 5 to 5.5) [43,49,50,53,73]. The presence of bridgmanite, as inferred from experiments, indicates a mantle in equilibrium with carbonate or carbon dioxide, the feedstock of life, but in disequilibria with hydrogen emanating from core, mantle and crust [31,32,69].

### 2.1. Olivine (Mg>Fe)_2_SiO_4_

This olive colored magnesium ferrous-iron silicate comprises much of the modern mafic and ultramafic upper mantle down to ~410 km, the basaltic ocean floor to our planet, numerous volcanic islands (e.g., the Galapagos), and in upthrusted (obducted) elements of ocean floor in mountain belts, the so-called ophiolites (e.g., Cyprus). Olivine also contains significant traces of water, nickel, manganese and, on occasion, pyrophosphate [74,75,76,77]. Both the nickel and manganese may be released to acidic hydrothermal solutions, while the pyrophosphate may have been released through high temperature acidic degassing [78]. However, mantle olivines appear to have low phosphate contents (typically 10 to ~60 ppm P, but reaching 0.2% on occasion) [79,80]. On land, soils developed over olivine-bearing basaltic rocks have been sought throughout the ages for their fertility [81,82]. And today basalts are even quarried and crushed as a general-purpose fertilizer and as a mitigating agent of anthropogenic CO_2_ increase [10,83].

In the Hadean (4.5 to 4.0 billion years before present (Ga)), in what is now the Earth’s mantle, olivine began to precipitate in a relatively short-lived magma ocean and in super plumes induced through giant impacts [8,70,84,85]. A depth of 410 km marks the beginning of a transition zone in the mantle. Under these pressures, the orthorhombic olivine crystals, with a density (ρ_0_) of ~3.3 g/cm^3^, give way to the compositionally equivalent, though denser spinel-structured ringwoodite (ρ_0_ = 3.65 g/cm^3^) [86,87], with the likely expulsion of trapped water at high pore pressure [88,89,90]. This pressure-induced mineral transition to ringwoodite is only a prelude to the much more significant ‘collapse’ of that spinel structure to a silicate perovskite crystal type with a density of ~4.2 g/cm^3^ [91]. The perovskite that comprises the lower mantle is designated ‘bridgmanite’, our next concern. This transition produces the sharp upper-to-lower mantle discontinuity at a depth of 660 km, broken on occasion by those giant impact-induced buoyant super plumes [15,84,90,92,93,94,95] (Figure 3).

### 2.2. Bridgmanite (Mg,Fe)SiO_3_

Bridgmanite, the silicate perovskite, is the most abundant mineral in the Earth, comprising ~80% of the lower mantle, with the remainder being made up of magnesiowüstite ([Mg,Fe]O) and Ca-perovskite (~Ca[Fe,Ti]O_3_) [92]. The lower mantle extends from the 660 km (23 GPa) discontinuity down to the outer core boundary at 2900 km (135 GPa) (Figure 1 and Figure 3) [57,95,102]. Yet its presence was only initially surmised from experimental work [93]. This is because bridgmanite is a very rare mineral, generally found only in meteorites where it is generated by transitory high shock waves pressures of 18 to 25 GPa [95,103,104,105].

So why is bridgmanite so central to astrobiology? The short answer is that its formation generates a mantle with an oxygen fugacity (i.e., an effective partial pressure) high enough to render much of the carbon into its most oxidized from, i.e., as carbon dioxide, the main substrate of life [1,54,106]. This CO_2_-producing chemistry is facilitated by the ‘collapse’ of the ringwoodite lattice structure to the relatively condensed perovskite lattice. Perovskite needs a metal ion with a valency of three to build its structure. This ion is generally Al^3+^, but there is an inadequate supply of aluminum in the upper mantle to satisfy this requirement, and none at all in ringwoodite that occupies a lower pressure zone just above the 660 km discontinuity. The need for a trivalent metal ion is satisfied by the forced disproportionation of the Fe^2+^ ions in the olivine/ringwoodite lattices to Fe^3+^ and Fe^0^ (metallic iron) [15,57,107]. While the Fe^3+^ fulfilled this need in the perovskite structure of bridgmanite, the newly orphaned high density Fe^0^ would have been excluded to the surrounds and much of it sunk to the core as the magma ocean cooled, leaving the mantle oxidized enough to be buffered by quartz-fayalite-magnetite (QFM). With iron being partly oxidized in magnetite (Fe^2+^Fe_2_^3+^O_4_), at mantle temperatures much of the carbon was forced into the 4+ oxidation state, i.e., as CO_2_ (Figure 1 and Figure 2) [5,48,55,92,100,108,109,110,111]. This carbon dioxide-producing physical chemistry facilitated by the perovskites produced an H_2_O- and CO_2_-dominated atmosphere that dates from the final stages of Earth’s accretion [15,112].

This theory whereby the surviving magnetite controls the effective partial pressure of CO_2_ has been put to various tests and investigations, notably by a demonstration that accounts for the ‘heavy’ iron isotope composition of the Earth’s basalts as compared to similar samples from Mars and the asteroid Vesta [113]. In fact, Martian basalts are generally much more reduced than those of Earth [114,115]. Indeed, judging from Martian meteorites, it appears that the redox state of the Martian mantle is buffered around, or even below, the iron-iron oxide (iron-wüstite, Fe:FeO) boundary [97]. And it is calculated that the core/mantle boundary in Mars is defined by ringwoodite (Mg,Fe)_2_SiO_4_) to solid iron [116,117,118]. In such a case gases evolved from the reduced Martian magmas and resulting from giant impacts would have mainly comprised hydrogen, water vapour, methane and carbon monoxide [8,16]. In fact, notwithstanding earlier arguments [119], as the olivine/ringwoodite-to-bridgmanite transition is not realized (which requires a pressure of 23 GPa to trigger the said transition), planets the size of Mars or smaller are much less likely to emit large quantities of CO_2_—the staple of life—through degassing and vulcanism [16,20,48,116].

### 2.3. Pyrrhotite Fe_(1-x)_S (x = 0 to 0.2)

Accompanying the olivine in the mantle was the partially oxidized (sulfidized) iron sulfide pyrrhotite, along with its nickel-bearing equivalent pentlandite ([Fe,Ni]_9_S_8_) [120]. It also occurs within, or more generally at the base of komatiitic lava flows, introduced as a sulfide melt along with the silicate lavas [121]. The mineral is vital as the source of sulfur dissolving in hydrothermal solutions as a required constituent of bioorganic molecules. During alkaline hydrothermal convection, some of the sulfur component of the pyrrhotite is released to solution as bisulfide (HS^–^), to be precipitated, on meeting with ferrous iron from the ocean as a minor FeS (mackinawite) component of the alkaline hydrothermal mound (Figure 1 and Figure 2) [3,39,46,122,123,124,125].

### 2.4. Serpentine (Mg,Fe,)_2-3_Si_2_O_5_(OH)_4_

If olivine is subjected to interaction with water at temperatures up to ~300 °C it hydrates exothermically in a process called serpentinization, the main thermal drive to the alkaline convective hydrothermal cells [126]. As the serpentine has a greater volume and lower density than olivine the effect is to force fine-scale cracking of the upper crust [127]. In the AVT the mechanism of serpentinization was invoked to predict the occurrence of submarine open system hydrothermal convection cells that would generate off-ridge alkaline springs [3,55,128]. These springs would feed the reducing agent hydrogen in the spring waters as a source of electrons to generate organic molecules in the resulting hydrothermal precipitate mound (Figure 1 and Figure 2) [4,52]. In this scenario the serpentinizing system is an ‘upstream’ mechanism or engine whose output feeds the hatchery of life in a submarine alkaline spring mound as electrons, methane and formate interact in the mineral membrane with carbon dioxide and nitrate and nitrite (Figure 2) [4,14,62,129,130].

However, Tutolo et al. [131] show that in the silica-rich Hadean ocean, minimal hydrogen would be generated at nearly two orders of magnitude lower than we had formerly assumed. Nevertheless, we do know that large quantities of hydrogen would have degassed from sources in the Earth’s mantle and from the mantle/core boundary [31,32,103,132,133,134]. Also, Tutolo et al. [131] did demonstrate that the pH contrast at the Hadean submarine alkaline—as an ambient proton motive force—to be up to two orders of magnitude greater than we had first surmised [4]. This is significant because Hudson and his collaborators [44] have shown that in a test of a version of AVT, the formic acid (HCOOH) produced from CO_2_ derives its hydrogen from external protons in the presence of hydrogen at 1.5 bars [44].

One geophysical difference between early Earth and now, is the likelihood that whereas the mantle was substantially hotter in the Hadean [34,135], the oceanic crust was stagnant prior to the onset of plate tectonics produced by volcanic over-, inter- and under-plating, fed from super plumes. Hence, at certain locations, the thermal gradient in dormant sections would have been much lower on early Earth [35,37]. At the same time radial and concentric fracturing produced by mantle bulges over and around super plume heads likely penetrated through the 40 to 80 km thickness of the crust. Moreover, “downward-excavating” open hydrothermal convective serpentinizing systems may have plumbed substantially deeper than they do today, leaching more methane [136,137], and lasting even longer that the ~10^5^ years estimated by Ludwig et al. [138]. As CO_2_ escaped from the mantle, it would be reduced to CH_4_ along the fracture walls at temperatures below ~400 °C, before the methane was entrained in the aforementioned hydrothermal cells [37,54,55,136]. This methane is another potential fuel, one that could have introduced abiotically reduced carbon to a putative metabolism [39,55,130,139,140,141,142] (Figure 1 and Figure 2).

We conclude that the disequilibria at the alkaline spring/acidulous ocean interface was theoretically sufficient to drive and nurture emergent life [38,39,129,130,142,143,144].

## 3. Two Minerals to Make Life Happen

In the presence of the disequilibria produced by the four minerals that set the stage for life’s emergence, two minerals, fougerite ([Fe^2+^_6x_Fe^3+^_6(x−1)_O_12_H_2(7−3x)_]^2+^·[(CO^2−^)·3H_2_O]^2−^) and mackinawite (Fe[Ni]S), are vital as initial “free energy” conductors and converters of such disequilibria, and are considered as the initiators of a CO_2_-reducing metabolism in a membrane constituted of these same minerals [43]. The iron-bearing oxyhydroxides and subordinate sulfide are precipitated at the ocean-crust interface where the alkaline hydrothermal fluids meet the acidulous ocean water bearing iron and other transition elements fed to the ocean from the ≤410 °C acidic springs [45,57,58]. But because the molarity of hydroxide in the alkaline fluids is so much higher than that of the sulfide, the oxyhydroxide fougerite (green rust) dominates the precipitates [56] (Figure 4) rather than iron sulfide as previously thought [4,43,145].

### 3.1. Fougerite [Fe^2+^_6x_Fe^3+^_6(x−1)_O_12_H_2(7−3__x)_]^2+^·[(CO^2−^)·3H_2_O]^2−^

The green rust mineral fougerite is a mixed-valence redox-flexible semi-conducting naturally-occurring anionic clay, dosed with Mg^2+^, Ni^2+^, Mn^2+^, and Co^2+^ [146,147]. Fougerite’s extensive inner surfaces appear to provide the ‘mechanistic’ potential to fill the roles of the redox- and pH-converter that enabled life’s emergence by driving endergonic—thermodynamically uphill—processes [41,43,62,130,145,148,149,150,151]. And there was certainly no want for fougerite in the all-enveloping early ocean—the mineral precursor to the diagenetic magnetite comprising the first known banded iron formation outcropping in western Greenland [45,59,60,152,153,154]. Mimicking this natural process of precipitation and transformation, Konstantinos Simeonidis and his collaborators [155] have generated green rust on a path to nanometric idiomorphic crystals of magnetite—a mineral with potential in catalysis, biotechnology and water remediation, though it is inimical to membrane formation. Their continuous processing mechanism employed nitrate ions to oxidize ferrous iron in aqueous solution as demonstrated by Hansen and collaborators [156] and adopted by Russell and colleagues [41,43] in the AVT (eqn 1 from Asimakidou et al., [155]):5 Fe(OH)_2_ + Fe^2+^ + SO_4_^2−^ + 0.25 NO_3_^−^ + (n +1.5) H_2_O →[Fe_4_^2+^,Fe_2_^3+^,(OH)_12_]^2+^·[SO_4_·nH_2_O]^2−^ + 0.25 NH_3_ + 0.25 OH^−^(1)

Following from demonstrations of the variable valence fougerite to act as an inorganic nitrate/nitrite reductase, Barge and her collaborators show that ammonium can aminate pyruvate (itself theoretically provided by hydrogenation of CO_2_ on the mineral greigite) [157] to the amino acid alanine in the presence of fougerite [157,158] (Figure 4 and Figure 5). Moreover, Tosca and collaborators [73] demonstrate the generation of hydrogen as green rust is oxidized by water which would provide another source of H_2_ at an alkaline vent. And Arrabito and collaborators [159] demonstrate a general consanguinity between life and green rust as they also draw attention to how the biocompatibility of the double layer hydroxides, including green rust, have been extensively exploited in the biomedical industry. Yet to be tested are (i) the presumed potential of green rust situated in the membrane to also act as a proton wire, a proton pyrophosphatase, methane monooxygenase, polymerase and (ii) as an engine of synthesis in the production of aromatic rings (cf. quinones and flavins) [62,65,130,145,150,151,160,161,162].

Computer simulations have provided insights that help in the planning of such experiments [159,168,169,170]. And the relatively recent development of operando techniques should allow demonstrations of, for example, coupling of steep redox/pH gradients along the metal and hydrous layers respectively including reductive recharge, to the driving of other endergonic reactions [171]. Moreover, there is some theoretical support for seeing the interior galleries of double layer hydroxides such as fougerite offering the beginnings of a guidance or information system. The first consideration in this respect is to enquire, and investigate, how fougerite might couple to a fluctuating or a varying environment beyond mere ‘static’ determinism [146,172,173,174]. The greater the asymmetric response or rates of response to reversals of the driving force the more impact this would have on evolution [172,175]. Reactions along such paths that lead, at each step, to a limited autonomy through the development of improved information systems should also result in a fuel-saving economy [172]. We turn next to the single layered sulfide, mackinawite, as a necessary support mineral.

### 3.2. Mackinawite Fe(Ni)S

Mackinawite—an electron conductor [176]—is the subordinate sulfide analogue of fougerite, precipitating with it at the submarine alkaline spring, and comprising a small inner portion of the membranes dividing ocean from alkaline hydrothermal solution (Figure 5) [56,177]. Although, like fougerite it is a layered mineral, Bourdoiseau and coworkers [178] have shown, against earlier expectations, that it includes no layered hydrous intercalations. However, mackinawite does retain its structure during partial oxidation, though not by the insertion of anions, but rather through the loss of Fe^2+^ to solution to maintain charge parity [178,179,180].

Nevertheless, like fougerite, mackinawite diadochically absorbs Ni^2+^, Mg^2+^, Mn^2+^, and Co^2+^ in the ferrous iron site [61,160,181,182,183,184]. Indeed, nickeliferous mackinawite also had a protometabolic role in AVT [3,61,185]. Demonstrating the first “crack” in the kinetic barrier to life, i.e., one that offered the ‘escape route’ to an autotrophic protometabolism, Hudson and his collaborators [44] demonstrated the role of an Fe(Ni)S precipitate in the effective hydrogenation of CO_2_ to formate through the application of a steep pH gradient—one of the predictions of the AVT [3,42,186] (Figure 5).

In the hydrothermal conditions at the vent, as with fougerite [73], mackinawite maybe oxidized/sulfidized to greigite with the release of hydrogen and/or its absorption as electrons and protons (cf. the ‘cubane’ active centers to the affine iron-sulfur enzymes) [61,187]. Whether the conformational oxidation to greigite can be reversed in this nanoworld scenario is an open question. If so the two conformations might support action as a disequilibria converter, a possibility that also invites further research [188]. Judging from density functional theory (DFT) calculations, greigite (Fe_3_S_4_) too has a further potential protometabolic role, that of catalyzing the hydrogenation of CO_2_—via the reduction of the carbonyl moiety of the intermediate glyoxylic acid (CHO-COOH)—to acetic and pyruvic acids [157,189].

## 4. The Relevance of Accretion Histories to Astrobiology

In the assessments of which worlds might meet the disequilibria requirements for the emergence of life, accretion histories are informative. Within our solar system the terrestrial planets were accreted mainly from iron-nickel meteorites and wet enstatite (MgSiO_3_) chondrites (ECs) [19,163,194,195]. In contrast, the outer planetary zone was populated with CCs comprising phosphoran (P_2_O_5_-bearing) olivine (where P substitutes for Si), serpentine and organic molecules [19,163,194,195]. The CCs represent the all-important contribution of carbon molecules for the emergence of life. However, they formed, along with larger outer solar system bodies, beyond the ‘snow line’ where water and other volatiles condense into ices [196]. While Jupiter’s deep gravitational well had the tendency to block the inward migration of these CCs, sufficient numbers did manage to slip through a gap in the disc as Jupiter got larger and migrated inwards [197]. This resulted in much of the outer asteroid belt also being populated with CCs [197]. Their inward migration may have been responsible for a late heavy bombardment of the inner solar system bodies, adding carbon to the terrestrial planets at the same time [198]. We should note in passing that the migration of Jupiter itself may have had a deleterious effect on Venus’ habitability as it forced the planet into high orbital eccentricities [199]. Such deviations may have driven water loss and brought about a runaway greenhouse notwithstanding the relatively low solar luminosity [199,200,201].

As we have noted, reduced carbon is ubiquitous in the outer solar system and the Universe at large, making it—in the absence of oxidants—a poor signature for life detection on its own. Even so, another poor life detection signature for terrestrial exoplanets throughout habitable zones in the galaxy are oxygen atmospheres, which may be abiotically-produced from extreme water loss due to high energy UV flux acting to continually disperse hydrogen from vaporizing oceans [202]. Yet, for life to be driven to emerge requires oxidants to provide a positive ‘electrode’ to the reduced molecules hydrothermally focused at a planet or moon’s exterior surface [14]. That is why we contend that the astrobiological signature of interest is the observation of terrestrial water worlds within habitable zones of sufficient mass to drive the physicochemical pressure and stresses such that the bridgmanite-dominant mineralogy of the mantle is poised around the quartz-fayalite-magnetite- (QFM, i.e., SiO_2_-Fe^2+^_2_SiO_4_-Fe^2+^Fe^3+^_2_O_4_) buffer. This control dictates CO_2_ as the stable state of carbon above ~400 °C and thereby a carbon dioxide-rich atmosphere [55].

## 5. Discussion

Although our planet is largely an amalgam of metal-bearing chondrites, some of them carbonaceous, the volatisphere (atmosphere and ocean) has been relatively oxidized over the last 4.4 Ga. Carbon occurs as the dioxide rather than hydride, sulfur as sulfide, polysulfide and sulfate, and nitrogen as N_2_, although it was also accompanied in the atmosphere by minor concentrations of nitrogen oxides that dissolve as nitrate and nitrite in the ocean [28,29,47]. While the latter relatively oxidized gases and ions are results of solar radiation, carbon dioxide has been a major component emanating from our planet since the solar wind blew off the earliest and ephemeral hydrogen atmosphere. This Hadean CO_2_ atmosphere was produced, as has been argued, through the oxidation of the lower mantle through disproportionation of ferrous iron in olivine/ringwoodite to produce the ferric-bearing perovskite, bridgmanite, in the lower mantle, while the abandoned native iron tended to exit the lower mantle as it gravitated toward the core, leaving CO_2_ as the stable but volatile state of carbon in the mantle. Of course, the Hadean was anything but an equable time in Earth’s history and we should expect there to have been major vacillations in the content and temperature of the volatisphere. However, given the 500-million-year duration of the Hadean era, vacillations were more than likely to have intersected the conditions that drove life into being much of this time. We might assume the same held for our sister planet Venus when young, whose atmosphere, without life‘s draw-down, now boasts ~90 bars of the gas [199].

The question then arises, just how deep does a core-mantle boundary have to be on a wet-rocky world to produce and hold a relatively oxidized atmosphere, i.e., to allow a bridgmanite-dominated lower mantle? The mantle-core boundary depth of Mars today, for example, appears not to reach the threshold of bridgmanite stability, leaving its supposed early atmospheric oxidation state uncertain at best [92,97,203,204,205]. However, Mars does presently have a CO_2_ atmosphere, although with an overall pressure amounting to only 12 mbar [206]. Carbonates too are sparse so whether it once had a higher CO_2_ pressure, as supposed by climate modelling, is also debatable [207,208,209]. More serious is the likelihood that a bridgmanite zone never existed and the Noachian atmosphere would have been H_2_ and H_2_O, both being highly soluble in ringwoodite [210,211,212] (Figure 3).

Nevertheless, all the other rocky, wet, and icy and bodies smaller than Mars are very unlikely to have oxidizing atmospheres—i.e., they would be devoid of those electron acceptors required by life—unless they have been and are being, subjected to intense radiation or have entertained large numbers of CO_2_-rich comets.

However, in terms of life’s emergence on our planet, we have seen that olivine, and its two “offspring”, bridgmanite and serpentine, and accompanying pyrrhotite, did set the stage for life’s emergence. Comprising a portion of a membrane separating alkaline hydrothermal fluid from carbonic ocean water, fougerite can act as a hydrogen producer, nitrate/nitrite to ammonium converter, an aminase and phosphate attractor and low entropy environment for its condensation to pyrophosphate (as in olivine and the layered mineral canaphite) [155,156,177,213,214,215]. At the same time, Ni-bearing iron sulfide, likely mackinawite, and its offspring Ni-bearing greigite, can act as a hydrogenase, hydrogenating catalyst, free energy converter and electron wire or conductor [44,216,217]. Indeed, the contribution by Hudson and collaborators demonstrates the power of a proton gradient (as carbonic acid) across an iron-nickel sulfide, probably mackinawite—an ambient proton motive force—to drive the hydrogenation of CO_2_ to the organic molecule, formate (Table 1) (Figure 5). To quote from Hudson et al. [44]; “overall our results suggest that H_2_ is the main electron donor, that a large pH gradient is necessary for its oxidation, and that sulfide is insufficient (and might not be required) as an electron donor.” Their breakthrough experiment finally answers Leduc’s [1] early plea to recognize that “(T)he most important problem of synthetic biology is...the reduction of carbonic acid”!

It may seem that our focus on just these six minerals is overly reductive. For example, haven’t clays been at the forefront of minerobiolization hypotheses and experiments since Bernal [148,192]? But green rust/fougerite is a clay [151] and of the other likely vent precipitates—hisingerite (Fe^3+^_2_Si_2_O_5_[OH]_4_·2H_2_O), greenalite (~[Fe^2+^,Fe^3+^]_2_3[Si_2_O_5_][OH]_4_), accompanied by amorphous silica—fougerite is the prime candidate [49,59,146,152]. It has that distinction owing to its variable valence sites, its propensity to juggle electrons and protons in and out of its extensive reactive and flexible internal surfaces, and its proven worth as an abiotic nitrate/nitrite reductase, aminase and generator of hydrogen fuel from water [146,156,158,218,219].

And what of the electron-rich metallic minerals such as awaruite [Ni_3_Fe] that Russell and collaborators [220] had originally called upon to act as a catalyst in the reduction of CO_2_ and CO in the serpentinizing ocean crust? Indeed, in some recent exciting milestone experiments Preiner and collaborators [221] detail how H_2_ and CO_2_ do react in the presence of awaruite (and magnetite, Fe_3_O_4_) to produce formate, acetate, pyruvate, methanol and methane—all the biochemical products of the acetyl coenzyme A pathway. Moreover, in whole rock two-feed flow reactor serpentinization experiments where the charges included olivine and the reduced iron mineral pyrrhotite White and coworkers [39] recorded formate, acetate and sporadic traces of methane.

On the strength of Preiner and coworkers’ one-pot incubation experiments, Martin [222] has argued that awaruite was the “hydrothermal vent alloy” that catalysed the acetyl coenzyme A pathway before the advent of genes—the “Square 1 of bacterial evolution” [222]. But the statement that “serpentinizing systems could have preceded and patterned biotic pathways” does not meet with Endre’s [223] stricture that “complexification can only take place in small steps” that produce ever higher efficiencies of entropy production [41,224,225]. As Nick Lane [226] puts it “life transcends chemistry” and is certainly not “a chemical reaction” nor can it be directly compared to a present-day industrial process [222,227].

But the idea that awaruite was a precursor catalyst was mooted at a time when Russell and Hall [4] had considered “the entire hydrothermal system as a pre-living entity and that evolution had brought about a miniaturization of scale from kilometres to millimetres and eventually to micrometres” at the vent [3,128,220,228]. In more recent formulations of the AVT the vent structure itself (where awaruite could not form) has been taken as the site of life’s conception; sown and succoured from crustal emanations while bathed in the Hadean ocean [130,229]. Under this view mackinawite and fougerite offer isolated but fixed nickel ions to the prebiotic system in Fe:Ni ratios more consistent with the low solubility of nickel in hydrothermal solutions [7,9,130,182,229,230,231,232]. Indeed, the very presence of “residual” awaruite in the oceanic crust is evidence for the low solubility of nickel relative to iron in serpentinizing systems [231,232].

Nevertheless, our focus on the relative concentrations on Earth of the six minerals detailed here should not lull us into a consideration of mere scalar processes. Indeed, a recurring topic in AVT is the vital role of directed active transport; from inward migration of materials in the solar disc, gravitationally-driven convective transport in and on the planet, to ionic, including proton transport or translocation and electron conduction through the membrane—a prelude to vectorial metabolism [233]. And at the alkaline hydrothermal vent itself, one might imagine the fate of the two nickel-bearing iron nanocrystals to be “drawn down” into the entropy-generating vortex of emerging life, dressed in their organic polymers—the precursors of the structure-function-conformational relationships that can still be discerned in life today [145,177,234,235].

## 6. Astrobiological Implications

Studies of the perovskite mineral bridgmanite [(Mg,Fe)SiO_3_], responsible for our oxidized mantle, indicate that ringwoodite suffers a sharp density-increasing phase change to bridgmanite at depth of ~660 km which corresponds to a pressure of 21 GPa. Thus, the original magma ocean and mafic to ultramafic volcanoes would have exhaled carbon mainly as carbon dioxide to the ocean floor, which produced an ocean and atmosphere comprising mostly electron acceptors rather than donors (Figure 1 and Figure 2) [8,55,236,237]. High potential oxidants were also likely available as nitrate and nitrite derived from lightning-driven oxidation of N_2_ in the CO_2_ atmosphere [28,238]. The reductants consisting of hydrogen and methane, were generated in the lower temperatures of the exothermically-serpentinizing, long reduction-path lengths of the open convective hydrothermal systems feeding the vents and precipitate mounds [55,129,136]. The juxtaposition of the relatively oxidizing and acidulous early Hadean ocean with highly reduced alkaline hydrothermal springs, resulted in the spontaneous precipitates of Ni-bearing, Fe oxyhydroxide and sulfide barriers comprising nanocrysts of fougerite and mackinawite. At times and in places, these barriers induced the necessary redox and pH gradients to force the reduction of CO_2_ [42,44], and possibly the oxidation of CH_4_, thus driving the first steps of an autotrophic metabolism [42,130].

However, the wet and rocky bodies in the solar system smaller than Earth and Venus probably have more reduced mantles comprising little or no bridgmanite. This consideration could explain the predominance of methane on the moons of Jupiter and Saturn [48,239,240,241,242,243,244] and perhaps even early Mars [203,204,205,245,246]

## 7. Caveats and Limitations

While the several approaches employed here appear to converge on the limiting mineral fundaments to life’s emergence, without which there would be no early CO_2_ oxidizing atmospheres and therefore no obvious electrochemical disequilibria required to drive life’s emergence, nor to sustain habitability, there are caveats:

(1) AVT assumes that phototrophy cannot emerge *de novo* but evolves via autotrophy and heterotrophy—a point of view open to debate and astrobiological exploration [14];

(2) The assumption derived from climate modelling that Mars once had a CO_2_ atmosphere with a pressure exceeding 250 mbar is at odds with the likely high hydrogen fugacity (i.e., the very low oxygen fugacity) of the Martian mantle and the paucity of carbonate outcrop. Although such an early atmosphere has not been disproven, high emissions of H_2_ could also exoplain the warm temperatures at that time [92,203,206,209,247,248];

(3) Some carbon dioxide feedstock on Mars and sub-Mars sized bodies that had no shortage of volatile reductant fuels, might have been provided, and continuously so, from CO_2_-bearing comets in our solar system [88,249];

(4) Oxidants on moons such as Europa may be generated qualitatively by relatively local high energy radiation [250,251,252];

(5) It is not possible to tell as yet whether the conclusions presented here are applicable to exoplanet exploration. The challenge is to demonstrate the availability of electron acceptors on these other worlds up to 1.6 Earth radii (e.g., [9,253,254]. Indeed, the conclusions reached in this essay would not necessarily apply to M dwarf systems where initial conditions of planetary formation and evolution might have been very different [204,255,256,257,258,259,260]. For example, ‘super-earths’ now in the ‘habitable zones’ of M dwarfs may have suffered irreversible runaway greenhouse conditions (cf. Venus). Later these would have been enveloped with abiotically-produced oxygen atmospheres resulting from a high energy UV flux sufficient to drive water loss from vaporizing oceans through the continual dispersion of H_2_ [202]. While such atmospheres are likely to be overwhelmed with oxygen gas and suffer atmospheric warming inimical to life, some smaller examples might have both a hydrosphere as well oxygen pressures suitable for O_2_ to act as a positive ‘electrode’ for life’s emergence, a possibility enhanced by the addition of nitric oxides produced by high energy coronal mass injections [29].

## Figures and Tables

**Figure 1 life-10-00291-f001:**
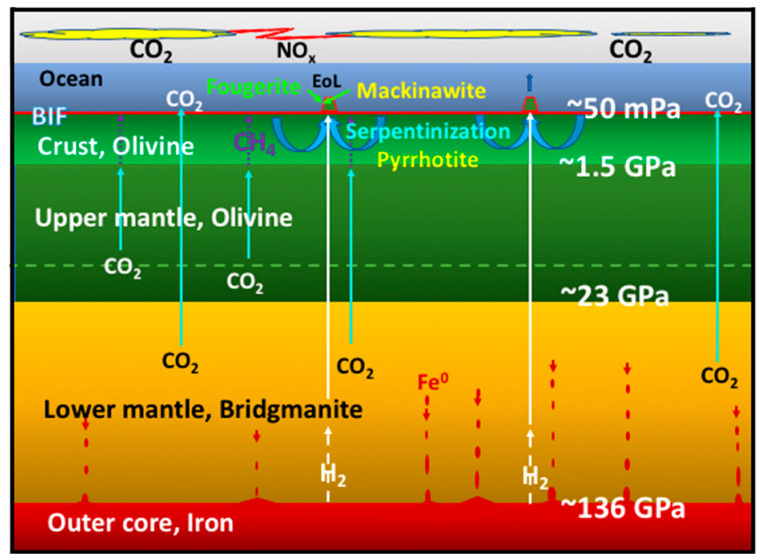
Cross-section sketch of Earth in the Hadean showing the mechanisms—the coupled engines—that promote the conditions for the emergence of life (EoL). The result is the generation of precipitate hydrothermal mounds comprising fougerite, subordinate mackinawite and silica gel (Figure 2) [39,45,46,47]. The diagram indicates the whereabouts of the six ‘must-have’ mineral phases discussed in the text. Pressure in excess of 23 GPa metamorphoses olivine and ringwoodite to the perovskite mineral, bridgmanite, leading to the disproportionation of ferrous iron to Fe^0^ and Fe^3+^ in the early magma ocean. The gravitation of the native iron toward the core left a mantle enriched in ferric iron buffered at quartz-fayalite-magnetite (QFM: SiO_2_-Fe^2+^_2_SiO_4_-Fe^2+^Fe^3+^_2_O_4_) [15,48]. CO_2_, degassing rapidly from olivine-bearing komatiitic lavas and intrusives fed from super-plumes from deep within the relatively oxidized hot mantle, bubbles up at high pressure onto the ocean floor, producing a carbonic ocean with extreme pH in relatively quiescent periods of between 5 and 5.5 units [35,49,50,51,52,53]. Cloud-to-cloud lightning produces NO_x_ that, when dissolved in the ocean, generates nitrate and nitrite ions [28]. Any CO_2_ trapped in the thick Hadean oceanic crust slowly converts to CH_4_ and formate below ~400 °C, some of which is entrained in the moderate temperature alkaline hydrothermal systems [37,39,46,54,55,56]. The partial dissolution of pyrrhotite and the hydration and oxidation of olivine to serpentine in the oceanic crust through convective circulation of ocean water generates the moderate temperature alkaline springs bearing HS^–^ [39,46]. These alkaline fluids also entrain H_2_ emanating from the mantle and core-mantle boundary [37,39]. Up to 80 mmol/liter of ferrous iron and subordinate transition metals derived from a myriad of ≤410 °C acidic springs (not shown) were pumped into the early acidulous Hadean ocean where much of the iron remained in supersaturated solution [45,57,58]. For the most part this iron was only precipitated as fougerite and subordinate mackinawite (accompanied by silica gel) on meeting submarine alkaline hydrothermal springs bearing ≤10 micromoles of the bisulfide (HS^–^) and 0.1 to 10 millimoles of the hydroxyl ion (OH^–^). The result was the generation of precipitate hydrothermal mounds comprising fougerite, subordinate mackinawite and silica gel [39,45,46,47]. Peripheral fougerite also gave rise to the Banded Iron Formations (BIF) [59,60]. In alkaline vent theory (AVT) one of the fougerite-mackinawite-silica mounds so precipitated on the ocean floor was the hatchery of life (EoL: Emergence of Life) [4,39,41,43,61].

**Figure 2 life-10-00291-f002:**
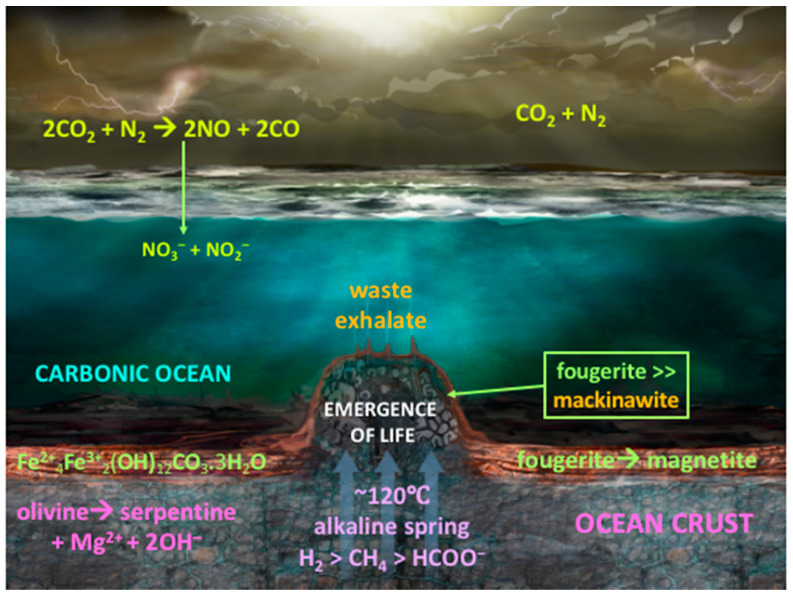
The Hadean Earth as a water world [41]. The ocean was carbonic and nitrate/nitrite-bearing, with pH in relatively quiescent periods between 5 and 5.5, having been injected from below with high pressure CO_2_ exhalations from the mantle; from solidifying komatiitic lavas and intrusive magmas in equilibrium with QFM, and from above with NOx produced by cloud-to-cloud lightning [15,28,51,53]. The partial dissolution of pyrrhotite and the hydration and oxidation of olivine to serpentine in the ~40–80 km thick Hadean oceanic crust generates the moderate temperature alkaline springs bearing HS^−^ as well as CH_4_ and formate produced in the crust by the reduction (hydrogenation) of CO_2_ [35,37,39,46,54,55]. It also entrains H_2_ emanating from the mantle and core-mantle boundary [31,32,69]. Transition metals derived from myriad ≤410 °C acidic springs (not shown) [45,57,58] and present in metastable state in the ocean, are spontaneously precipitated as fougerite and subordinate mackinawite on meeting the alkaline hot springs to produce the hydrothermal mound argued to be the hatchery of life as in Figure 1 (EoL) [4,39,41,43,61].

**Figure 3 life-10-00291-f003:**
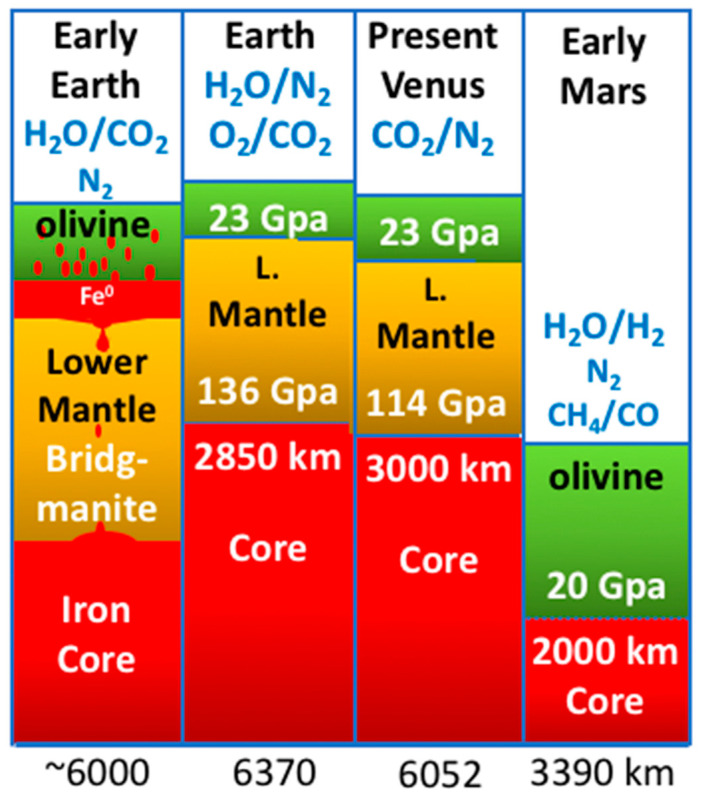
Depth and pressure comparisons between the early Earth (with native iron gravitating to the core), and present-day Earth, Venus and Mars. Bridgmanite is produced at pressures of ~21–23 Gpa and marks the upper to lower mantle (ringwoodite to bridgmanite) transition zone on Earth and Venus, a pressure barely reached on Mars. It follows that present-day Earth has a mantle at QFM, as does Venus. Impact-induced mantle super plumes were also responsible for the partial homogenization of the mantles of Earth and probably Venus [84,96]. The Martian mantle is likely to be far more reduced, hydrogen-buffered around iron-wüstite (Fe:FeO) [97]. Assuming magma oceans occupied portions of all three planets, it follows that the atmospheres of the early Earth and probably early Venus comprised CO_2_ and H_2_O, whereas the early atmosphere/hydrosphere of Mars consisted of H_2_ + H_2_O > CH_4_ + CO. A percentage of N_2_ is common to all [98]. Figure based on [5,16,20,99,100,101].

**Figure 4 life-10-00291-f004:**
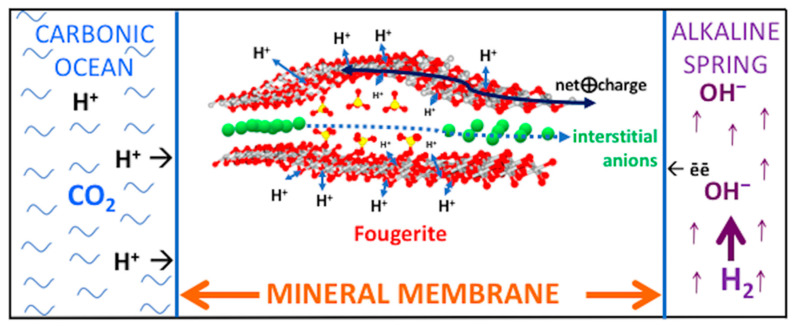
Cross-section of two individual iron oxyhydroxide layers of the double layered mineral fougerite. In the AVT fougerite nano- to micro-crystals comprise the mid and outer portions of an inorganic membrane (see Figure 5) precipitated by—but separating the acidulous ocean from—the alkaline hydrothermal spring waters [145,150,151]. The detailed structure shows the contrasting heights of the interstitial space (the interlayer) between two layers of a redox-flexible fougerite nanocrystal (e.g., Fe^2+^_4_Fe^3+^_2_[OH]_12_·CO_3_·3H_2_O ⇔ Fe^2+^_2_Fe^3+^_4_O_2_[OH]_10_·CO_3_·3H_2_O) (Figure 5). Intercalated between these inner surfaces are spherical ions (e.g., chloride and/or carbonate) forcing a gallery height of 0.75 nm, and tetragonal ions (e.g., sulfate and/or condensed phosphate, P_2_O_5_ as in olivine) which expand the height to ~1.1 nm [163]. Stresses associated with such conformational flexuring may be measured in piconewtons, comparable to those operating in the motor protein myosin [164,165,166]. Any motion of charge along the Fe-oxyhydroxide layers will be accompanied by modifications in the p*K* values of the OH groups such that more oxidized Fe^3+^ sites will tend to deprotonate their ambient hydroxy groups, thus releasing H^+^ into the interstices [167]. Drift of Fe^3+^ sites has the potential to pull interstitial anions along the galleries to produce condensations and other reactions in this low entropy environment. In this scenario the fougerite mineral acts as a pump or nanoengine. For example, as the electron current is drawn toward the oxidants, so the 3+ charge on the iron atoms would migrate in the opposite direction with the potential to drag carboxylic anions from the outer periphery inwards to react with ammonium formed in the same structure and, therefrom, synthesize amino acids [43,156,158]. Electrons will hop in directions counter to the drift of the Fe^3+^ sites, while the hydrous interlayers could act as a proton wire whereby transport is facilitated by the grotthuss mechanism [162]. Note that the model nanocryst is just one of a myriad comprising the mineral membrane (cf. Figure 5).

**Figure 5 life-10-00291-f005:**
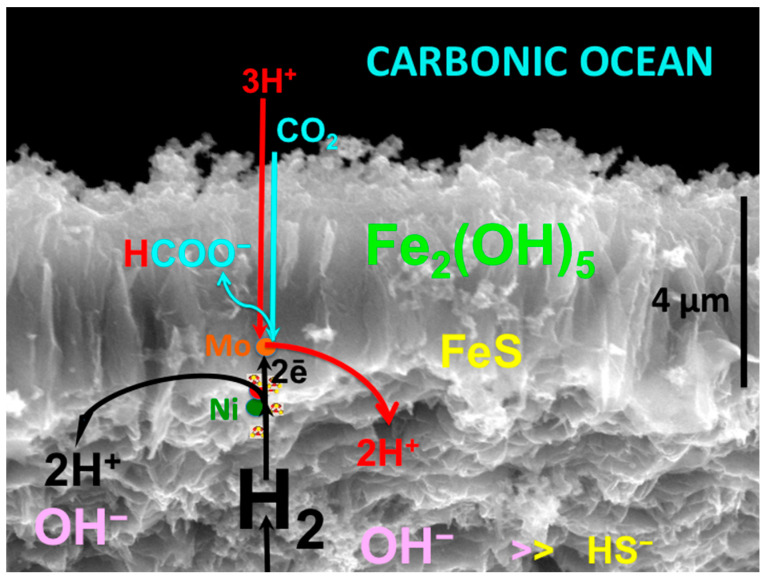
A putative reduction of CO_2_ in the Hadean ocean (top) to formate (HCOO^−^) in a membrane comprising the ‘must-have’ nickel-bearing minerals fougerite (~Fe_2_[OH]_5_) (central and outer zone) and subordinate mackinawite (FeS) (inner zone), dosed with molybdenum as further catalyst [56,190]. The alkaline hydrothermal solution occupies the bottom of the diagram. Formate generation is driven by protons from the ocean accessing the Ni-(and Mo-) dosed FeS nanocrystals through the outer fougerite layer [9,42,44,56,190,191]. In this speculative rendition, the formate is discharged into the hydrous interlayers of fougerite, a mineral known for mediating prebiotic biochemistry-like production (Table 1) [156,177]. To be compared to Hudson et al. [44] Figures 1A, 3D and S28.

**Table 1 life-10-00291-t001:** The six ‘must-have’ Hadean minerals as they are thought to contribute to the emergence of life on Earth.

Mineral	Contribution and Consequence	References
Olivine	Upper mantle/crust: precursor to bridgmanite & serpentine	[74,75,76,77,79,80]
Bridgmanite	Lower mantle mineral produced by metamorphism of Fe^2+^/Mg-silicate so forcing disproportionation of the Fe^2+^ as bridgmanite purloins Fe^3+^, effectively oxidizing the lower mantle as the orphaned Fe^0^ gravitates to the core	[92,95,103,105,107]
Pyrrhotite	Source of bisulfide (HS^−^) in the alkaline hydrothermal solutions	[3,39,46,120,121,123,124,125,192]
Serpentine	Hydration of olivine generates highly alkaline submarine springs with pH contrast with Hadean ocean of ~6 pH units	[3,4,38,39,55,126,127,128,129,130,131,136,137,142,143,144]
Fougerite	Dominant precipitate at vent, sufficiently complex as a membrane to have acted as embryonic life’s first disequilibria converter (as a general reductase, aminase, and possibly a polymerase and pyrophosphatase), H_2_ generator and proton transfer wire	[41,43,59,60,73,146,147,150,151,152,155,156,159,167,193]
Mackinawite	Subsidiary mineral acting as hydrogenase and electron wire	[42,56,61,160,177,178,179,180,181,182,184]

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
