# Peer review of "Six ‘Must-Have’ Minerals for Life’s Emergence: Olivine, Pyrrhotite, Bridgmanite, Serpentine, Fougerite and Mackinawite"

_life, 2020, doi:10.3390/life10110291_

Round 1
Reviewer 1 Report
The authors describe six minerals namely olivine, pyrrhotite, bridgmanite, serpentine, fougerite, and mackinawite, and discuss roles of these minerals for emergence of terrestrial life. They conclude that these minerals in alkaline hydrothermal vent system are essential for emergence of life. Overall, this manuscript is well-written, and I recommend for publication after minor modifications.
I am convinced that these six minerals have played essential roles in emergence of terrestrial life. However, the authors claimed that these minerals as “must-have”, I expect that the authors comment on possibilities (or impossbilities) of substitutes for these minerals (or system). It would be useful to considering habitability of other planets and exoplanets.
Minor points:
- Figs 4 and 5 can be efficiently managed to clarify their correspondence.
- Line 355 “Within our solar system the terrestrial planets were accreted...” requires references. I think some of [19, 161, 188-190] in line 358 should be moved after the above sentence.
Author Response
Response to referee
Referee 1
The authors describe six minerals namely olivine, pyrrhotite, bridgmanite, serpentine, fougerite, and mackinawite, and discuss roles of these minerals for emergence of terrestrial life. They conclude that these minerals in alkaline hydrothermal vent system are essential for emergence of life. Overall, this manuscript is well-written, and I recommend for publication after minor modifications.
Reply: Thank you.
I am convinced that these six minerals have played essential roles in emergence of terrestrial life. However, the authors claimed that these minerals as “must-have”, I expect that the authors comment on possibilities (or impossbilities) of substitutes for these minerals (or system). It would be useful to considering habitability of other planets and exoplanets.
Reply: This is a really good point and we now critically assess clays), awaruite (Ni3Fe) and greigite (Fe3S4) as substitutes for mackinawite and fougerite as suggested by Preiner et al. (2020) and Martin (2020) as introduced in the two paragraphs lines 485-493 and lines 494-522. Other references have been added to these paragraphs as deemed necessary for understanding the background to these new additions
Minor points:
- Figs 4 and 5 can be efficiently managed to clarify their correspondence.
Reply: We have now clarified the relationship in lines 313-316
- Line 355 “Within our solar system the terrestrial planets were accreted...” requires references. I think some of [19, 161, 188-190] in line 358 should be moved after the above sentence.
Reply: We thank the referee for pointing this disjuncture out which we have now corrected (new line 406
Reviewer 2 Report
The paper promoting the crucial role of six minerals looks very interesting and convincing.
Concerning the crucial role of minerals, it should be fair to quote the contribution of Hazen who reviewed the role of mineral surfaces in: R. M. Hazen, D. A. Sverjensky, Cold Spring Harbor Perspect. Biol. 2010, 2, a002162.
As for the hydrothermal environment, the authors should quote Westall F, Hickman-Lewis K, Hinman N et al. (2018) A Hydrothermal-Sedimentary Context for the Origin of Life. Astrobiology 18:259-293.
Putting quotation marks in a title is probably not very attractive. What about replacing “must-have” by mandatory?
Author Response
Referee 2
The paper promoting the crucial role of six minerals looks very interesting and convincing.
Reply: Thank you.
Concerning the crucial role of minerals, it should be fair to quote the contribution of Hazen who reviewed the role of mineral surfaces in: R. M. Hazen, D. A. Sverjensky, Cold Spring Harbor Perspect. Biol. 2010, 2, a002162.
Reply: With respect we consider doing so would confuse the reader given that not one of the 6 minerals promoted here are mentioned by Hazen et al. (2010). For the significance of mineral surfaces we already pay “homage” to the preceding authors: Goldschmidt 1952 [105], Bernal et al. 1960 [146], Arrhenius et al., 1993 [59], and Arrhenius, 2003 [150].
Nevertheless, in lieu of Hazen we now have added references MacKay (1960) and Cairns-Smith et al. (1992).
As for the hydrothermal environment, the authors should quote Westall F, Hickman-Lewis K, Hinman N et al. (2018) A Hydrothermal-Sedimentary Context for the Origin of Life. Astrobiology 18:259-293.
Reply: Again, with respect, the 31 year old “alkaline vent theory” – the basis of the present manuscript – stands in strong contrast to Francis Westall’s “sedimentary hydrothermal environment/exotic prebiotic molecule hypothesis” which lacks nanometric focus. Moreover, only one of the six minerals we propose is called upon, viz., olivine, but even so it is not considered it in the same senses as we do. Also, their favored precipitate, silica gel, unlike fougerite and mackinawite, offers no redox-cum-conformational responses such as to be expected of pumps and nanoengines required to enable life’s onset. And their argument regarding fluid dynamics has been refuted by Wang et al. 2019 [213] and Hudson et al. 2020 [44]. And, a last point, “diversity” spells contingencies rather than the low entropy environment required for emergent structure (e.g., the Big Bang).
Putting quotation marks in a title is probably not very attractive. What about replacing “must-have” by mandatory?
Reply: Whether something is “attractive” or not, depends on the eye of the beholder. To us, “mandatory” seems too severe a word for use in the developing transdisciplinary field of Astrobiology.